# Associations of combined phenotypic aging and genetic risk with incident cancer: A prospective cohort study

Lijun Bian[1,2†], Zhimin Ma[1†], Xiangjin Fu[1], Chen Ji[1], Tianpei Wang[1], Caiwang Yan[1,2,3], Juncheng Dai[1,2], Hongxia Ma[1,2], Zhibin Hu[1,2], Hongbing Shen[1,2,4]*, Lu Wang[3]*, Meng Zhu[1,2,3]*, Guangfu Jin[1,2,3]*

[1]Department of Epidemiology, Center for Global Health, School of Public Health, Nanjing Medical University, Nanjing, China; [2]Jiangsu Key Lab of Cancer Biomarkers, Prevention and Treatment, Collaborative Innovation Center for Cancer Personalized Medicine and China International Cooperation Center for Environment and Human Health Nanjing Medical University, Nanjing, China; [3]Department of Chronic Non-Communicable Disease Control, The Affiliated Wuxi Center for Disease Control and Prevention of Nanjing Medical University, Wuxi Center for Disease Control and Prevention, Wuxi Medical Center, Nanjing Medical University, Wuxi, China; [4]Research Units of Cohort Study on Cardiovascular Diseases and Cancers, Chinese Academy of Medical Sciences, Beijing, China

*For correspondence:
hbshen@njmu.edu.cn (HS);
wanglu123@njmu.edu.cn (LW);
zhmnjmu@njmu.edu.cn (MZ);
guangfujin@njmu.edu.cn (GJ)

†These authors contributed equally to this work

Competing interest: The authors declare that no competing interests exist.

## Abstract

**Background:** Age is the most important risk factor for cancer, but aging rates are heterogeneous across individuals. We explored a new measure of aging-Phenotypic Age (PhenoAge)-in the risk prediction of site-specific and overall cancer.

**Methods:** Using Cox regression models, we examined the association of Phenotypic Age Acceleration (PhenoAgeAccel) with cancer incidence by genetic risk group among 374,463 participants from the UK Biobank. We generated PhenoAge using chronological age and nine biomarkers, PhenoAgeAccel after subtracting the effect of chronological age by regression residual, and an incidence-weighted overall cancer polygenic risk score (CPRS) based on 20 cancer site-specific polygenic risk scores (PRSs).

**Results:** Compared with biologically younger participants, those older had a significantly higher risk of overall cancer, with hazard ratios (HRs) of 1.22 (95% confidence interval, 1.18–1.27) in men, and 1.26 (1.22–1.31) in women, respectively. A joint effect of genetic risk and PhenoAgeAccel was observed on overall cancer risk, with HRs of 2.29 (2.10–2.51) for men and 1.94 (1.78–2.11) for women with high genetic risk and older PhenoAge compared with those with low genetic risk and younger PhenoAge. PhenoAgeAccel was negatively associated with the number of healthy lifestyle factors (Beta = −1.01 in men, p<0.001; Beta = −0.98 in women, p<0.001).

**Conclusions:** Within and across genetic risk groups, older PhenoAge was consistently related to an increased risk of incident cancer with adjustment for chronological age and the aging process could be retarded by adherence to a healthy lifestyle.

**Funding:** This work was supported by the National Natural Science Foundation of China (82230110, 82125033, 82388102 to GJ; 82273714 to MZ); and the Excellent Youth Foundation of Jiangsu Province (BK20220100 to MZ).

## eLife assessment

This study presents **fundamental** findings that advance our understanding of the role of phenotypic aging in cancer risk. This article presents **compelling** results that show Phenotypic Age Acceleration (PhenoAgeAccel) can predict cancer incidence of different types and could be used with genetic risk to facilitate the identification of cancer-susceptible individuals. These results will be of broad interest to the research community and clinicians.

## Introduction

Cancer continues to be the leading cause of death globally and the reduction of cancer-related deaths remains to be a public health priority (*Bray et al., 2018*). The morbidity and mortality of cancer increase dramatically with age, which demonstrates that aging is the greatest risk factor for cancer (*Siegel et al., 2018*). Although everyone gets older, individuals are aging at different rates (*Rutledge et al., 2022*). Therefore, the variation in the pace of aging between person may reflect the differences in susceptibility to cancer and death. Thus, measurement of an individual's biological age, particularly at the early stage of life, may promote the primary and secondary prevention of cancer through earlier identification of high-risk groups.

Recently, Morgan and colleagues developed and validated a novel multi-system-based aging measurement (*Levine et al., 2018*), PhenoAge, which has been shown to capture long-term vulnerability to diseases like COVID-19, and strongly predict morbidity and mortality risk in diverse populations (*Kuo et al., 2021b*; *Liu et al., 2018*). However, it is largely unknown whether PhenoAge can predict overall cancer risk and identify high-risk individuals for potential personalized prevention.

To date, more than 2000 genetic loci have been identified as susceptibility markers for certain cancers by genome-wide association studies (GWAS) (*Buniello et al., 2019*). Although the effect of these individual loci is relatively modest on cancer risk, a PRS combining multiple loci together as an indicator of genetic risk has been proved to effectively predict the incidence of site-specific cancer (*Dai et al., 2019*; *Lecarpentier et al., 2017*; *Mars et al., 2020*). Recently, we systematically created site-specific cancer PRS for 20 cancer types, and constructed an incidence-weighted CPRS to assess the effect of genetic risk on overall incident cancer risk based on the UK Biobank (*Zhu et al., 2021*). Previous studies had indicated an interaction between genetic factors and age on cancer risk (*Mavaddat et al., 2015*). However, the extent to interaction between genetic factors and PhenoAge on overall cancer risk remained unclear.

In this study, we calculated PhenoAge in accordance with the method described previously and then evaluated the effectiveness of PhenoAge in predicting the risk of overall cancer in the UK Biobank. We also assessed the extent to which a level of accelerated aging was associated with an increased overall cancer risk across groups with a different genetic risk defined by the CPRS.

## Methods

### Participants

The participants included in this study are derived from the UK Biobank. The UK Biobank is a large population-based cohort study and detail protocol is publicly available (*Bycroft et al., 2018*). In brief, approximately 500,000 participants aged 40–70 were recruited from 22 assessment centers across England, Scotland, and Wales between 2006 and 2010 at baseline. Each eligible participant completed a written informed consent form and provided information on lifestyle and other potentially health-related aspects through extensive baseline questionnaires, interviews, and physical measurements. Meanwhile, biological samples of participants were also collected for biomarker assays and a blood draw was collected for genotyping. The UK Biobank study has approval from the Multi-center Research Ethics Committee, the National Information Governance Board for Health and Social Care in England and Wales, and the Community Health Index Advisory Group in Scotland (http://www.ukbiobank.ac.uk/ethics/).

### PhenoAge and PhenoAgeAccel PRS

We calculated PhenoAge in accordance with the method described previously (*Levine et al., 2018*). Briefly, PhenoAge was calculated based on mortality scores from the Gompertz proportional hazard model on chronological age and nine multi-system clinical chemistry biomarkers (albumin, creatinine,

**eLife digest** Age is a major risk factor for cancer. Other factors, such as lifestyle or environmental exposures, may increase or mitigate cancer risks. Biological age, which considers accelerated aging processes, may, however, better predict cancer risk than chronological age. Some scientists propose using biological aging measures as an alternative for assessing cancer and other age-related disease risks, as these markers may provide a more accurate assessment of the various factors contributing to cancer risk.

PhenoAge, a measure of biological aging processes in the body, could provide an alternative way to assessing aging-related cancer risks. This tool utilizes an individual's chronological age and nine biomarkers of aging processes. It has the potential to identify individuals whose aging process is accelerated compared to their peers, potentially indicating an increased cancer risk. This information may empower them to make lifestyle changes that could significantly reduce their risk.

To assess the suitability of PhenoAge, Bian, Ma et al. used nine clinical chemistry biomarkers and chronological age to calculate PhenoAge in 374,463 participants from the UK Biobank. Their findings revealed that people with older PhenoAges – regardless of their genetic risk profiles – have an increased risk of cancer. Individuals with higher PhenoAge scores, indicating accelerated biological aging, had a roughly 25 percent higher risk of developing cancer. Individuals with both a high genetic risk and higher PhenoAge score had roughly double the risk of cancer. People with lower PhenoAges were more likely to have healthier lifestyles. These results suggest that adopting healthier lifestyles may slow the aging process and reduce cancer risk.

While the analyses conducted by Bian, Ma et al. provide promising insights, they also underscore the need for further research. PhenoAge may offer a way to assess biological aging and identify individuals at higher risk of cancer. Those with higher PhenoAge scores may benefit from earlier cancer screening, and adopting a healthier lifestyle could potentially slow down the aging process and reduce their cancer risk. However, more studies in more diverse cohorts of people are needed to confirm that PhenoAge is a reliable marker for cancer risk and to test interventions to slow aging and reduce cancer risks in individuals with accelerated aging.

---

glucose, [log] C-reactive protein [CRP], lymphocyte percent, mean cell volume, red blood cell distribution width, alkaline phosphatase, and white blood cell count) to predict all-cause mortality. The Biomarkers in the UK Biobank were measured at baseline (2006–2010) for all participants. To correct distribution skewness, we set the top and bottom 1% of values to the 99th and first percentiles. The formula of PhenoAge is given by

$$PhenoAge = 141.50 + \frac{ln\left\{(-0.00553) \times \frac{(-1.51714) \times exp(xb)}{0.0076927}\right\}}{0.09165}$$

where

$$xb = -19.907 - 0.0336 \times albumin + 0.0095 \times creatinine + 0.1953 \times glucose + 0.0954 \times ln(CRP) - 0.0120 \times lymphocytepercentage + 0.0268 \times meancorpuscularvolume + 0.3306 \times redbloodcelldistributionwidth(RDW) + 0.00188 \times alkalinephosphatase + 0.0554 \times whitebloodcellcount + 0.0804 \times age$$

Finally, we calculated Phenotypic Age Acceleration (PhenoAgeAccel), which was defined as the residual resulting from a linear model when regressing Phenotypic Age on chronological age. Therefore, PhenoAgeAccel represents Phenotypic Age after accounting for chronological age (i.e. whether a person appears older [positive value] or younger [negative value] than expected, biologically, based on his/her age).

55 independent PhenoAgeAccel-associated SNPs ($p<5 \times 10^{-8}$) and corresponding effect sizes were derived from a large-scale PhenoAgeAccel GWAS including 107,460 individuals of European ancestry (**Kuo et al., 2021a**). A PhenoAgeAccel PRS was created using an additive model as previously

described (*Dai et al., 2019*). In short, the genotype dosage of each risk allele for each individual was summed after multiplying by its respective effect size of PhenoAgeAccel.

## PRS calculation and CPRS construction

The calculation of site-specific cancer PRSs and the construction of CPRSs have been described in our previous published study (*Zhu et al., 2021*). In brief, for individual cancer, risk-associated single nucleotide polymorphisms (SNPs) and corresponding effect sizes were derived from the largest published GWASs in terms of sample size. Next, the dosage of each risk allele for each individual was summed after multiplication with its respective effect size of site-specific cancer. Except for nonmelanoma skin cancer and those without relevant GWAS or significant genetic loci published by now, we derived PRSs for 20 cancer types in this analysis. To generate an indicator of genetic risk for overall cancer, we constructed the CPRS as follows:

$$CPRS_i = \sum_{k=1}^{K} h_k PRS_{i,k}$$

Where $CPRS_i$ is the cancer polygenic risk score of $i^{th}$ individual, $h_k$ is the age-standardized incidence of site-specific cancer $k$ in the UK population, and $PRS_{i,k}$ is the aforementioned PRS of site-specific cancer $k$. Given the different spectrum of cancer incidence between men and women, CPRS were constructed for males and females, respectively.

## Assessment of healthy lifestyle

We adopted five healthy lifestyle factors according to the World Cancer Research Fund/American Institute of Cancer Research recommendations (https://www.aicr.org/cancer-prevention/) (*Shams-White et al., 2019*), i.e., no current smoking, no alcohol consumption, regular physical activity, moderate BMI (body-mass index, 18.5~30), and a healthy diet pattern. Participants of no current smoking were defined as never smokers or former smokers who had quit smoking at least 30 years. No alcohol consumption was defined as never alcohol use. Regular physical activity was defined as at least 75 min of vigorous activity per week or 150 min of moderate activity per week (or an equivalent combination) or engaging in vigorous activity once and moderate physical activity at least 5 days a week (*Lourida et al., 2019*). A healthy diet pattern was ascertained consumption of an increased amount of fruits, vegetables, whole grains, fish, and a reduced amount of red meats and processed meats (*Lourida et al., 2019*). The lifestyle index ranged from 0 to 5, with a higher index indicating a healthier lifestyle.

## Outcomes

Outcomes of incident cancer events in the UK Biobank were ascertained through record electronic linkage with the National Health Service central registers and death registries in England, Wales, and Scotland. Complete follow-up was updated to 31 October 2015 for Scotland, and to 31 March 2016 for England & Wales. Cancer events were coded using the tenth Revision of the International Classification of Diseases. The outcome of all cancer events were obtained from data fields 40006 and 40005 of the UK Biobank.

## Statistical analysis

Cancer risk of participants in the UK Biobank was assessed from baseline until to the date of diagnosis, death, loss to follow-up, or date of complete follow-up, whichever occurred first. Multivariable Cox proportional hazards regression analyses were performed to assess associations between PhenoAgeAccel and cancer risk and to estimate hazard ratios (HRs) as well as 95% confidence intervals (CI). Schoenfeld residuals and log-log inspection were used to test the assumption of proportional hazards. HRs associated with per 5 years increased of PhenoAgeAccel was calculated for site-specific cancer and overall cancer, respectively. We compared HRs between biologically younger and older participants. In addition, we calculated HRs for participants at low (the bottom quintile of PhenoAgeAccel), intermediate (quintiles 2–4), and high (the top quintile) accelerated aging, and HRs for participants splitted by decile of accelerated aging.

Meanwhile, participants were also divided into low (the bottom quintile of CPRS), intermediate (quintiles 2–4), and high (the top quintile) genetic risk groups. Absolute risk within each subgroup

defined by PhenoAgeAccel and CPRS were calculated as the percentage of incident cancer cases occurring in a given group. We calculated absolute risk increase as the difference in cancer incidences among given groups, extrapolated the difference in 5 year event rates among given groups. The 95% CIs for the absolute risk increase were derived by drawing 1,000 bootstrap samples from the estimation dataset. We performed additive interaction analysis between genetic risk (defined by CPRS) and PhenoAgeAccel on overall cancer risk, as well as genetic risk (defined by PhenoAgeAccel PRS) and lifestyle on PhenoAgeAccel using two indexes: the relative excess risk due to interaction (RERI) and the attributable proportion due to interaction (AP) (*Li and Chambless, 2007*). The 95% CIs of the RERI and AP were estimated by bootstrap (n=5000), which would contain 0 if there was no additive interaction. We also used multivariable linear regression models to assess associations between the PhenoAgeAccel and individual lifestyle factors with adjustment for age, family history of cancer, Townsend deprivation index, height, and the first 10 principal components of ancestry. All the above-mentioned analyses were performed for men and women separately.

Participants with missing data on any of the covariates were multiple imputed, and independent analyses were also performed based on complete data for sensitivity analyses. Besides, to examine the reliability of our results, we conducted several sensitivity analyses: (1) reclassifying PhenoAgeAccel levels based on quartiles (bottom, 2–3, and top quartiles defined as low, intermediate, and high accelerated aging, respectively) or tertiles (corresponding to low, intermediate, and high accelerated aging) of PhenoAgeAccel; (2) reevaluating the effect of PhenoAgeAccel based on participants of unrelated British ancestry; (3) excluding incident cases of any cancer occurring during the two years of follow-up; and (4) retrained PhenoAge in cancer-free participants based on mortality. All p-values were two-sided and p<0.05 was considered statistically significant. All statistical analyses were performed with R software, version 3.6.3 (R Project for Statistical Computing).

## Results

### Participants

After removing participants who had withdrawn their consent, had been diagnosed with cancer before baseline, failed to be genotyped, reported a mismatch sex with genetic data, or with missing data on PhenoAge, the final analytic dataset included 374,463 eligible participants (173,431 men and 201,032 women) (*Figure 1*). Of which, 169,573 participants were biologically older, with 92,189 men, and 77,384 women, whose median PhenoAgeAccel were 3.28 (interquartile range [IQR]: 1.50–6.06) and 3.07 (IQR: 1.33–5.79), respectively; 204,890 participants were biologically younger, with 81,242 men and 123,648 women, whose median PhenoAgeAccel were –2.61 (IQR: –4.35––1.25) and –3.55 (IQR: –5.64––1.81), respectively (*Table 1*).

### Associations of PhenoAgeAccel with cancer risk

There were 22,370 incident cancer cases, with 11,532 men and 10,838 women, during a median follow-up of 7.09 years (IQR: 6.35–7.72). The PhenoAgeAccel was significantly associated with increased risk for cancer sites of lip-oral cavity-pharynx, esophagus, stomach, colon-rectum, pancreas, lung, breast, cervix uteri, corpus uteri, prostate, kidney, bladder, multiple myeloma, Hodgkin's disease, and lymphoid leukemia, while negatively associated with risk of prostate cancer after adjusting for chronological age and other covariates (*Figure 2*, *Supplementary file 1a*).

For overall cancer, we observed an obviously higher distribution of PhenoAgeAccel in incident cancer cases than participants without incident cancer in both men and women (*Figure 3A and B*). PhenoAgeAccel was significantly associated with an increased risk of overall cancer in men (HR = 1.15, 95% CI, 1.13–1.17, p<0.0001) and women (HR = 1.15, 95% CI, 1.13–1.17, p<0.0001) per 5 years increase (*Table 2*). We also observed a significant gradient increase in incident cancer risk from decile 1 to decile 10 of PhenoAgeAccel (*Figure 3C and D*). Compared with biologically younger participants, those older had a significantly higher risk of overall cancer, with HRs of 1.22 (95% CI, 1.18–1.27, p<0.0001) in men, 1.26 (95% CI, 1.22–1.31, p<0.0001) in women, respectively (*Figure 3E and F*). Besides, Compared with individuals at low accelerated aging (the bottom quintile of PhenoAgeAccel), those in the intermediate (quintiles 2–4) and high accelerated aging (the top quintile) had a significantly higher risk of overall cancer, with HRs of 1.15 (95% CI, 1.09–1.21, p<0.0001) and 1.44 (95% CI, 1.36–1.53, p<0.0001) in men, 1.15 (95% CI, 1.09–1.21, p<0.0001), and 1.46 (95% CI, 1.38–1.55,

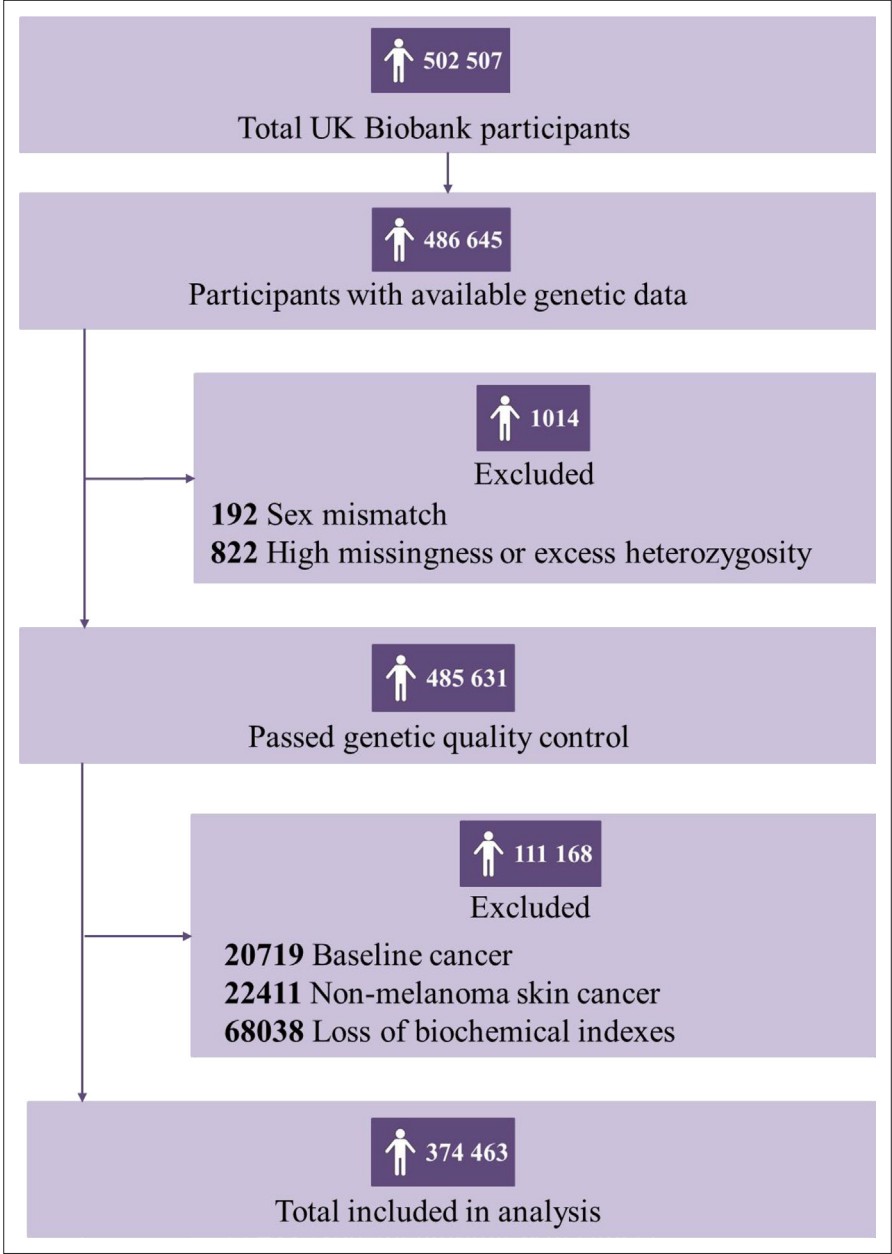

**Figure 1.** Flowchart for filtering participants from the UK Biobank cohort. Participants who had withdrawn their consent, had been diagnosed with cancer before baseline, failed to be genotyped, reported a mismatch sex with genetic data, or with missing data on Phenotypic Age (PhenoAge) were excluded.

p<0.0001) in women, respectively. These results did not change after adjustment for genetic risk and lifestyle factors (*Table 2*). Similar patterns were noted in a series of sensitivity analyses with reclassifying accelerated aging levels according to quartiles or tertiles of the PhenoAgeAccel (*Supplementary file 1b*), exclusion of incident cancer cases occurred during the two years of follow-up (*Supplementary file 1c*), in the unimputed data (*Supplementary file 1d*), in the unrelated British population (*Supplementary file 1e*), or using retrained PhenoAge in cancer-free participants (*Supplementary file 1f*).

## Joint effect and interaction of genetic factor and PhenoAgeAccel on overall cancer risk

The overall incident cancer risk is associated with both genetic risk and PhenoAgeAccel in a dose-response manner (*Figure 4*). Of participants with high genetic risk and older PhenoAge, the incidence

**Table 1.** Baseline characteristics of participants stratified by Phenotypic Age Acceleration (PhenoAgeAccel) categories.

| | Men (n=173,431) | | Women (n=201,032) | |
| --- | --- | --- | --- | --- |
| | Biologically younger n=81,242 | Biologically older n=92,189 | Biologically younger n=123,648 | Biologically older n=77,384 |
| PhenoAgeAccel, median (IQR), years | −2.61 (−4.35−−1.25) | 3.28 (1.50−6.06) | −3.55 (−5.64−−1.81) | 3.07 (1.33−5.79) |
| Age at baseline, median (IQR), years | 57.00 (49.00−62.00) | 58.00 (50.00−64.00) | 58.00 (51.00−63.00) | 56.00 (48.00−62.00) |
| Height, median (IQR), centimeters | 176.00 (171.00−181.00) | 175.00 (171.00−180.00) | 162.50 (158.00−167.00) | 162.00 (158.00−166.50) |
| Townsend deprivation index, median (IQR) | −2.34 (−3.76−0.16) | −1.86 (−3.52−1.10) | −2.33 (−3.73−0.07) | −1.79 (−3.44−1.10) |
| Family history of cancer, n (%) | | | | |
| No | 53907 (66.35) | 60654 (65.79) | 79800 (64.54) | 50924 (65.81) |
| Yes | 27335 (33.65) | 31535 (34.21) | 43848 (35.46) | 26460 (34.19) |
| Healthy lifestyle factors, n (%) | | | | |
| No current smoking | 46412 (57.13) | 43216 (46.88) | 78743 (63.68) | 45416 (58.69) |
| No alcohol consumption | 2064 (2.54) | 2828 (3.07) | 6221 (5.03) | 5459 (7.05) |
| Normal BMI | 66120 (81.39) | 62654 (67.96) | 103054 (83.34) | 49003 (63.32) |
| Regular physical activity | 54346 (66.89) | 57733 (62.62) | 80629 (65.21) | 46468 (60.05) |
| Healthy diet | 18540 (22.82) | 15486 (16.80) | 39547 (31.98) | 19826 (25.62) |
| Healthy lifestyle*, n (%) | | | | |
| Favorable | 7781 (9.58) | 5255 (5.70) | 17781 (14.38) | 7178 (9.28) |
| Intermediate | 57683 (71.00) | 57489 (62.36) | 87272 (70.58) | 49324 (63.74) |
| Unfavorable | 15778 (19.42) | 29445 (31.94) | 18595 (15.04) | 20882 (26.98) |
| Albumin, median (IQR), (g/L) | 46.20 (44.62−47.85) | 44.95 (43.28−46.64) | 45.47 (43.87−47.12) | 44.04 (42.42−45.70) |
| Alkaline phosphatase, median (IQR), (U/L) | 75.00 (64.30−87.40) | 82.80 (70.10−97.90) | 78.70 (65.10−93.90) | 86.20 (70.60−104.00) |
| Creatinine, median (IQR), (umol/L) | 77.60 (70.90−84.70) | 82.50 (74.40−91.60) | 61.70 (56.00−67.90) | 65.50 (58.90−73.20) |
| Glucose, median (IQR), (mmol/l) | 4.82 (4.49−5.14) | 5.09 (4.72−5.60) | 4.84 (4.54−5.14) | 5.05 (4.70−5.55) |
| C-reactive protein, median (IQR), (mg/dL) | 0.09 (0.05−0.16) | 0.19 (0.10−0.36) | 0.10 (0.05−0.20) | 0.25 (0.12−0.51) |
| Lymphocyte percent, median (IQR), (%) | 29.86 (25.48−34.48) | 25.60 (21.22−30.24) | 31.09 (26.58−35.80) | 27.00 (22.62−31.61) |
| Mean cell volume, median (IQR), (fL) | 90.90 (88.46−93.30) | 91.90 (89.10−94.74) | 91.10 (88.60−93.54) | 90.98 (87.64−94.02) |
| Red cell distribution width, median (IQR), (%) | 13.01 (12.70−13.37) | 13.63 (13.21−14.11) | 13.10 (12.71−13.50) | 13.90 (13.40−14.55) |
| White blood cell count, median (IQR), (1000 cells/uL) | 6.12 (5.29−7.10) | 7.27 (6.17−8.51) | 6.24 (5.37−7.26) | 7.41 (6.30−8.73) |

*Healthy lifestyle was defined as favorable (4–5 healthy lifestyle factors), intermediate (2–3 healthy lifestyle factors), and unfavorable (0–1 healthy lifestyle factor).

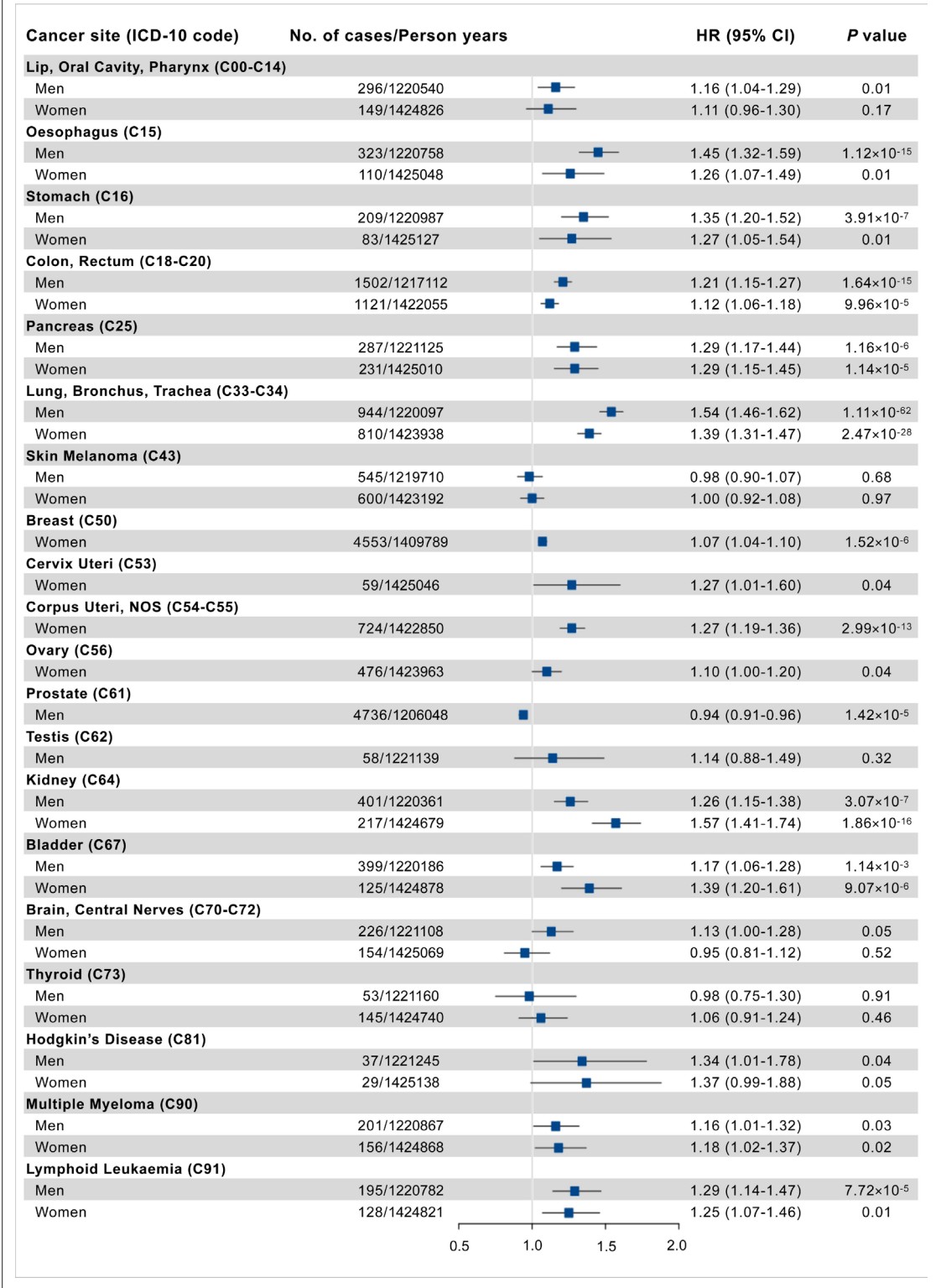

**Figure 2.** Association results of Phenotypic Age Acceleration (PhenoAgeAccel) with site-specific cancer risk per 5 years increased. Cox proportional hazards regression adjusted for age, height, cancer family history, Townsend deprivation index at recruitment, and the first 10 principal components of ancestry. Error bars are 95% confidence intervals (CIs).

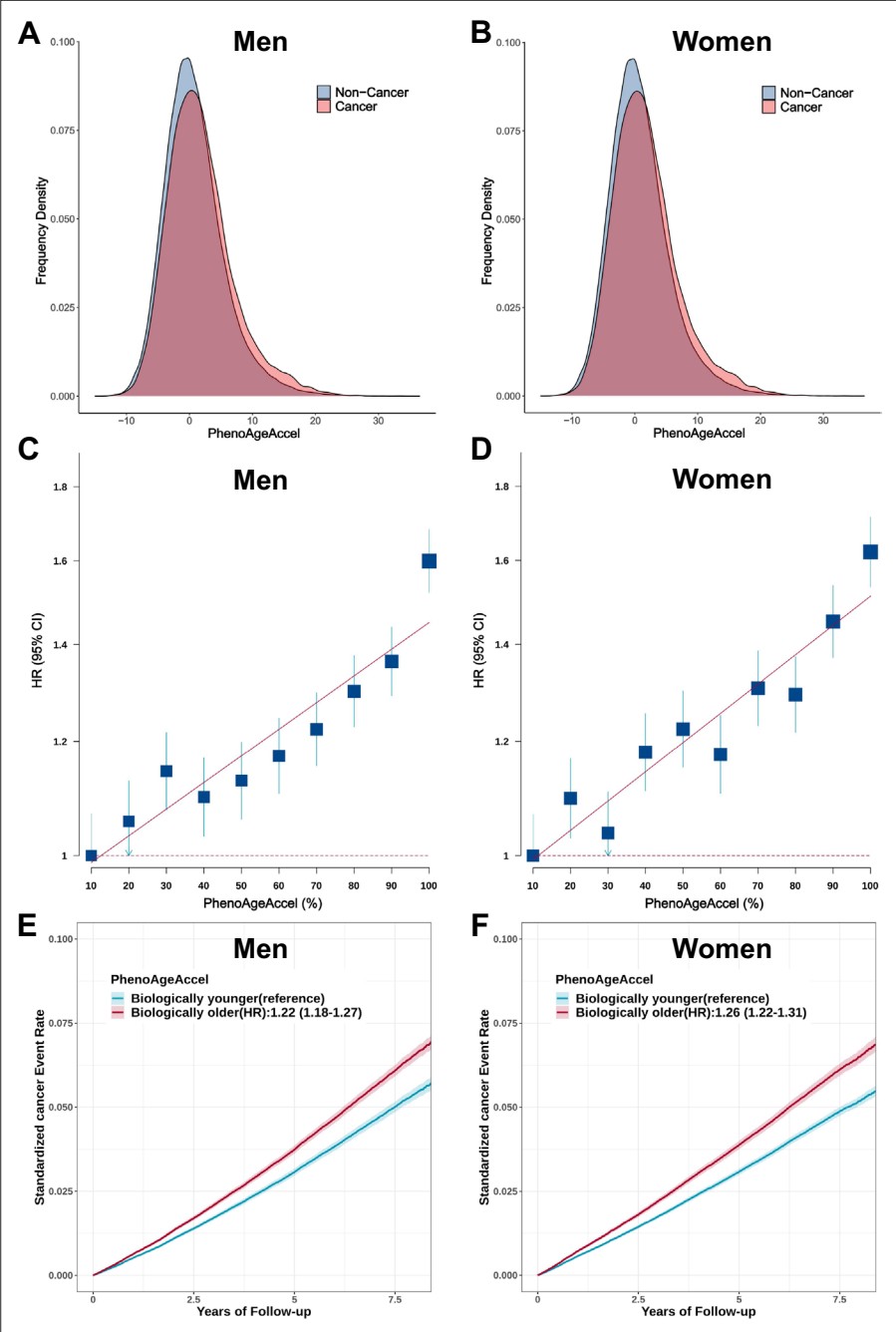

**Figure 3.** Effect of Phenotypic Age Acceleration (PhenoAgeAccel) on the risk of incident cancer in the UK Biobank. The distribution of PhenoAgeAccel between participants with incident cancer and those without incident cancer in the UK Biobank for men(A) and women (**B**). Participants in the UK Biobank were divided into ten equal groups according to the PhenoAgeAccel for men (**C**) and women (**D**), and the hazard ratios (HRs) of each group were compared with those in the bottom decile of PhenoAgeAccel. Error bars are 95% confidence intervals (CIs). Standardized rates of cancer events in younger and older PhenoAge groups in the UK Biobank for men (**E**) and women (**F**). HRs and 95% CIs were estimated using Cox proportional hazard models with adjustment for age, height, family history of cancer, Townsend deprivation index, and the first 10 principal components of ancestry. Shaded areas are 95% CIs.

**Table 2.** Association between Phenotypic Age Acceleration (PhenoAgeAccel) and cancer risk.

| | n (cases) | Person-Years | Model 1* | | Model 2† | |
|---|---|---|---|---|---|---|
| | | | HR (95% CI) | p value | HR (95% CI) | value |
| **Men** | | | | | | |
| **Per 5 years increase** | 173431 (11532) | 1190628 | 1.15 (1.13–1.17) | $1.55 \times 10^{-54}$ | 1.13 (1.11–1.15) | $5.46 \times 10^{-44}$ |
| **Category** | | | | | | |
| Biologically younger | 81242 (4649) | 564818 | Ref. | | Ref. | |
| Biologically older | 92189 (6883) | 625810 | 1.22 (1.18–1.27) | $5.31 \times 10^{-26}$ | 1.19 (1.15–1.24) | $3.18 \times 10^{-20}$ |
| **Quintiles†** | | | | | | |
| Low | 34687 (1864) | 242519 | Ref. | | Ref. | |
| Intermediate | 104058 (6668) | 716312 | 1.15 (1.09–1.21) | $2.17 \times 10^{-7}$ | 1.12 (1.07–1.18) | $8.01 \times 10^{-6}$ |
| High | 34686 (3000) | 231797 | 1.44 (1.36–1.53) | $1.01 \times 10^{-34}$ | 1.39 (1.31–1.47) | $1.99 \times 10^{-27}$ |
| ***p for trend*** | | | $3.03 \times 10^{-37}$ | | $1.65 \times 10^{-29}$ | |
| **Women** | | | | | | |
| **Per 5 years increase** | 201032 (10838) | 1393562 | 1.15 (1.13–1.17) | $1.80 \times 10^{-53}$ | 1.13 (1.11–1.15) | $1.35 \times 10^{-41}$ |
| **Category** | | | | | | |
| Biologically younger | 123648 (6229) | 861570 | Ref. | | Ref. | |
| Biologically older | 77384 (4609) | 531993 | 1.26 (1.22–1.31) | $5.91 \times 10^{-33}$ | 1.23 (1.18–1.28) | $2.71 \times 10^{-25}$ |
| **Quintiles‡** | | | | | | |
| Low | 40207 (1946) | 281912 | Ref. | | Ref. | |
| Intermediate | 120618 (6351) | 836341 | 1.15 (1.09–1.21) | $1.46 \times 10^{-7}$ | 1.13 (1.07–1.18) | $5.89 \times 10^{-6}$ |
| High | 40207 (2541) | 275309 | 1.46 (1.38–1.55) | $2.54 \times 10^{-36}$ | 1.40 (1.32–1.49) | $5.94 \times 10^{-28}$ |
| ***p for trend*** | | | $4.23 \times 10^{-37}$ | | $1.36 \times 10^{-28}$ | |

*Cox proportional hazards regression adjusted for Model 1, as well as cancer polygenic risk score and healthy lifestyle.

†Defined by quintiles of PhenoAgeAccel: low (the bottom quintile), intermediate (quintiles 2–4), and high (the top quintile).

‡Cox proportional hazards regression adjusted for age, height, cancer family history, Townsend deprivation index at recruitment, and the first 10 principal components of ancestry.

CI = confidence interval. HR = hazards ratio. Ref = reference.

**A**

| Subgroup | No. of cases/ Total no. | Incidence/ 100,000 py | | HR (95% CI) | *P* value |
|---|---|---|---|---|---|
| **Low genetic risk** | | | | | |
| Biologically younger | 668/16458 | 581.06 | | Reference | Reference |
| Biologically older | 1031/18229 | 827.94 | | 1.30 (1.18-1.43) | $1.22 \times 10^{-7}$ |
| **Intermediate genetic risk** | | | | | |
| Biologically younger | 2606/48636 | 768.85 | | 1.29 (1.18-1.40) | $6.45 \times 10^{-9}$ |
| Biologically older | 4012/55421 | 1064.81 | | 1.64 (1.51-1.78) | $1.02 \times 10^{-31}$ |
| **High genetic risk** | | | | | |
| Biologically younger | 1375/16148 | 1239.78 | | 2.11 (1.92-2.31) | $1.09 \times 10^{-55}$ |
| Biologically older | 1840/18539 | 1477.89 | | 2.29 (2.10-2.51) | $1.33 \times 10^{-74}$ |
| | | | 0.5   1.0   2.0   4.0 | | |

**B**

| Subgroup | No. of cases/ Total no. | Incidence/ 100,000 py | | HR (95% CI) | *P* value |
|---|---|---|---|---|---|
| **Low genetic risk** | | | | | |
| Biologically younger | 1030/24736 | 594.71 | | Reference | Reference |
| Biologically older | 761/15471 | 711.64 | | 1.25(1.14-1.38) | $2.14 \times 10^{-6}$ |
| **Intermediate genetic risk** | | | | | |
| Biologically younger | 3579/74310 | 690.11 | | 1.17(1.09-1.25) | $9.55 \times 10^{-6}$ |
| Biologically older | 2704/46308 | 848.31 | | 1.52(1.41-1.63) | $4.61 \times 10^{-30}$ |
| **High genetic risk** | | | | | |
| Biologically younger | 1620/24602 | 954.27 | | 1.63(1.51-1.77) | $1.03 \times 10^{-34}$ |
| Biologically older | 1144/15605 | 1076.17 | | 1.94(1.78-2.11) | $1.85 \times 10^{-53}$ |
| | | | 0.5   1.0   2.0   4.0 | | |

**Figure 4.** Risk of incident cancer according to genetic and Phenotypic Age Acceleration (PhenoAgeAccel) categories in the UK Biobank for men (**A**) and women (**B**). The hazard ratios (HRs) were estimated using Cox proportional hazard models with adjustment for age, height, family history of cancer, Townsend deprivation index, and the first 10 principal components of ancestry. Participants were divided into younger and older PhenoAge under different genetic risk groups. Error bars are 95% confidence intervals (CIs).

The online version of this article includes the following figure supplement(s) for figure 4:

**Figure supplement 1.** Risk of incident cancer according to genetic and Phenotypic Age Acceleration (PhenoAgeAccel) categories (quintiles) in the UKB cohort for men (**A**) and women (**B**).

rates of overall cancer per 100,000 person-years were estimated to be 1477.89 (95% CI, 1410.87–1544.92) in men and 1076.17 (95% CI, 1014.14–1138.19) in women versus 581.06 (95% CI, 537.12–625.00) in men and 594.71 (95% CI, 558.50–630.92) in women with low genetic risk and younger PhenoAge. Approximate double risks [HR, 2.29 (95% CI, 2.10–2.51) in men, p<0.0001; 1.94 (95% CI, 1.78–2.11) in women, p<0.0001] were observed in participants with high genetic risk and older PhenoAge, compared with those with low genetic risk and younger PhenoAge. Similar patterns were noted by reclassifying accelerated aging levels into low (the bottom quintile of PhenoAgeAccel), intermediate (quintiles 2–4), and high (the top quintile) (*Figure 4—figure supplement 1*). However, we did not observe the interaction between genetic and PhenoAgeAccel on overall cancer risk in men and women (*Supplementary file 1g*).

**Table 3.** Risk of incident cancer according to Phenotypic Age Acceleration (PhenoAgeAccel) categories within each genetic risk level[*].

| Gender | PhenoAgeAccel category | Low genetic risk | | Intermediate genetic risk | | High genetic risk | |
| --- | --- | --- | --- | --- | --- | --- | --- |
| | | Biologically younger | Biologically older | Biologically younger | Biologically older | Biologically younger | Biologically older |
| Men | No. of cases/Person years | 668/114962 | 1031/124526 | 2606/338950 | 4012/376782 | 1375/110907 | 1840/124502 |
| | Hazards ratio (95% CI) | Ref. | 1.29 (1.17–1.42) | Ref. | 1.27 (1.21–1.33) | Ref. | 1.10 (1.02–1.18) |
| | p value | | $3.53 \times 10^{-7}$ | | $3.15 \times 10^{-21}$ | | $1.07 \times 10^{-2}$ |
| | Absolute risk (%)- 5 years (95% CI) | 2.71 (2.49–2.94) | 3.87 (3.60–4.14) | 3.57 (3.41–3.72) | 4.95 (4.78–5.13) | 5.78 (5.44–6.11) | 6.90 (6.55–7.26) |
| | Absolute risk increase (%)- 5 years (95% CI) | Ref. | 1.16 (0.84–1.46) | Ref. | 1.39 (1.19–1.60) | Ref. | 1.13 (0.68–1.52) |
| Women | No. of cases/Person years | 1030/173195 | 761/106937 | 3579/518611 | 2704/318753 | 1620/169764 | 1144/106303 |
| | Hazards ratio (95% CI) | Ref. | 1.25 (1.14–1.38) | Ref. | 1.30 (1.24–1.37) | Ref. | 1.19 (1.10–1.28) |
| | p value | | $2.65 \times 10^{-6}$ | | $1.84 \times 10^{-24}$ | | $8.19 \times 10^{-6}$ |
| | Absolute risk (%) - 5 years (95% CI) | 2.83 (2.63–3.02) | 3.39 (3.12–3.65) | 3.27 (3.15–3.39) | 4.02 (3.86–4.19) | 4.58 (4.33–4.83) | 5.17 (4.84–5.49) |
| | Absolute risk increase (%)- 5 years (95% CI) | Ref. | 0.56 (0.26–0.86) | Ref. | 0.76 (0.57–0.93) | Ref. | 0.59 (0.23–0.98) |

[*]Cox proportional hazards regression is adjusted for age, height, family history of cancer, Townsend deprivation index, height, and the first 10 principal components of ancestry.

CI = confidence interval. Ref = reference.

## Disadvantages of older PhenoAge with overall incident cancer

In further stratification analyses by genetic risk category with younger PhenoAge as the reference group, we confirmed that older PhenoAge was significantly associated with a higher incident cancer risk across genetic risk groups (*Table 3*). Among participants at high genetic risk, the standardized 5 year incident cancer rates were 5.78% and 4.58% for biologically younger men and women versus 6.90% and 5.17% for those older, respectively. Similarly, among participants at low genetic risk, the standardized 5 year incident cancer rates increased from 2.71% and 2.83% for biologically younger to 3.87% and 3.39% for those older in men and women, respectively. Similar patterns were noted by reclassifying accelerated aging levels into low (the bottom quintile of PhenoAgeAccel), intermediate (quintiles 2–4), and high (the top quintile) (*Supplementary file 1h*).

In addition, to evaluate the implication for cancer screening in populations with different PhenoAgeAccel, we estimated the 5 year absolute risk of overall cancer between biologically younger and older participants with increasing age. Assuming 2% of absolute risk within the next 5 years as the threshold to be recommended for cancer screening, biologically younger men would reach the threshold at age 52, whereas those older men would reach the threshold at age 50; similarly, biologically younger women would reach the 2% of 5 year absolute risk at age 46, whereas those older women would reach the threshold at age 44 (*Figure 5*).

## Associations of lifestyle factors with PhenoAgeAccel

In the UK Biobank, biologically younger men (9.6%, 7781/81,242) and women (14.4%, 17,781/123,648) were more likely to have a favorable lifestyle (4–5 healthy lifestyle factors) than older men (5.7%, 5255/92,189) and women (9.3%, 7178/77,384) (*Table 1*). Among both men and women, we observed that PhenoAgeAccel decreased with the increase of healthy lifestyle factors (*Supplementary file 1i*).

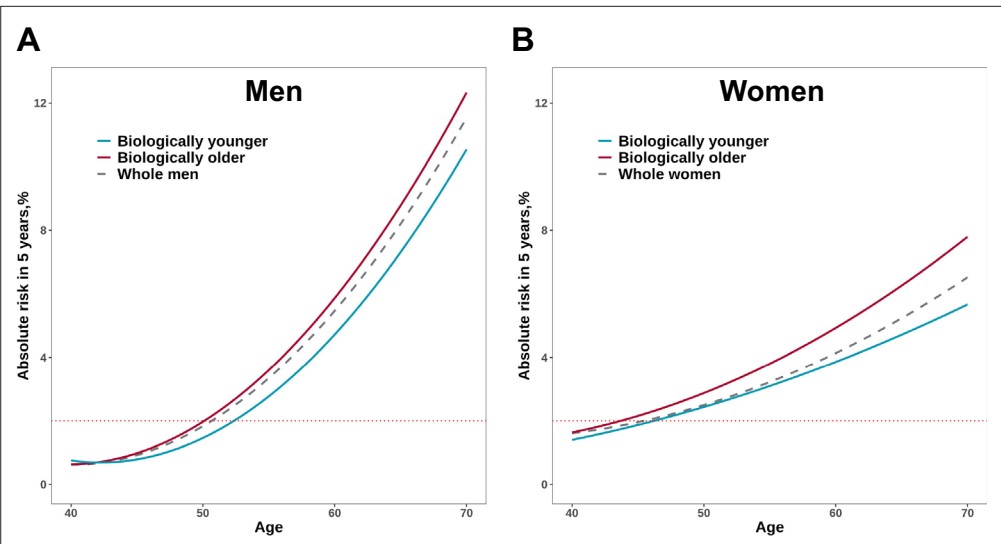

**Figure 5.** Absolute risk estimates of overall cancer based on the UK Biobank for men (**A**) and women (**B**). The x-axis is chronological age. The curves describe the average risk of participants in younger and older Phenotypic Age (PhenoAge) groups. The dashed curve represents the average risk of the whole population at different ages. The red horizontal dotted line represents 2% of 5 year absolute risks of overall cancer.

In addition, we found a significant negative correlation between the number of healthy lifestyle factors and PhenoAgeAccel (Beta = –1.01 in men, p<0.001; Beta = –0.98 in women, p<0.001) (**Supplementary file 1j**). However, we did not observe any interaction between genetic risk and lifestyle on PhenoAgeAccel in both men and women (**Supplementary file 1k**).

## Discussion

In this study, we calculated PhenoAgeAccel to explore the effect of accelerated aging on the risk of cancer, and demonstrated a positive association between accelerated aging and increased cancer risk after adjustment for chronological age in the UK Biobank. Meanwhile, older PhenoAge was consistently associated with an increased absolute risk of incident cancer within each genetic risk group; and participants with high genetic risk and older PhenoAge had the greatest incident cancer risk. Therefore, our findings provided the evidence for PhenoAgeAccel to be used for risk stratification of cancer, which were independent from genetic risk. Moreover, we also demonstrated that participants with older biological age often reaches the screening threshold 2 years in advance compared with biologically younger peers; and keeping a healthy lifestyle can effectively slow down the aging process.

Older age has been long recognized as the main risk factor for cancer, and the multistage model of carcinogenesis posits that the exponential increase in cancer incidence with age were mainly resulted from the sequential accumulation of oncogenic mutations in different tissues throughout life (**Laconi et al., 2020**). In consistent with this, age and exposure (i.e. smoking, ultraviolet light) dependent mutation signatures have been identified in several cancers by tissue sequencing (**Alexandrov et al., 2020**). However, biological aging is an enormously complex process and is thought to be influenced by multiple genetic and environmental factors (**van Dongen et al., 2016**). Therefore, several biomarkers, i.e., 'aging clocks' derived from epigenomic, transcriptomic, proteomic, and metabolomic data, have been proposed to measure the biological age and predict the risk of cancer and other diseases (**Rutledge et al., 2022**; **Zhang et al., 2022**). However, these measures were usually based on omics data and was not suitable for application in large populations by now. As a result, results from this study would provide a cost-effective indicator for measuring biological age as well as a novel biomarker for cancer risk prediction.

The associations between biological age and cancer risk has been investigated by several studies recently (**Li et al., 2022**) explored three DNA methylation phenotypic age and cancer risk in four subsets of a population-based cohort from Germany, and reported strong positive associations for lung cancer, while strong inverse associations for breast cancer (**Li et al., 2022**).

Meanwhile, results from the Melbourne Collaborative Cohort Study reported that epigenetic aging was associated with increased cancer risk of kidney cancer and B-cell lymphoma (*Dugué et al., 2018*). However, because of sample size, the association results were still inconsistent for DNA methylation phenotypic age among different studies. Leukocyte telomere length was also significantly associated with age and were regarded as an indicator of aging. Based on data from the UK Biobank, (*Schneider et al., 2022*) recently explored the associations between telomere length and risk of several diseases, and reported significant positive associations of telomere length for lymphoid leukemia, multiple myeloma, non-Hodgkin lymphoma, esophagus cancer, while negative associations for malignant neoplasm of brain, mesothelioma, and melanoma (*Schneider et al., 2022*). The positive associations were in consistent with our findings, however, the negative associations were not significant in our study. Meanwhile, the study did not indicate associations for other cancers, including cancers of lung, stomach, pancreas, and kidney, which showed relatively large effects (HR >1.3) in our study. These findings indicated that the different measures of biological age may reflect the different aspects of aging, and could be joint application in cancer risk assessment.

Recently, several studies have confirmed the associations between PhenoAgeAccel and cancer risk. *Mak et al., 2023* explored three measures of biological age, including PhenoAge, and assessed their associations with the incidence of overall cancer and five common cancers (breast, prostate, lung, colorectal, and melanoma). In our previous study, we investigated the association between PhenoAgeAccel and lung cancer risk and analyzed the joint and interactive effects of PhenoAgeAccel and genetic factors on the risk of lung cancer (*Ma et al., 2023*). In comparison to these studies, our analysis expanded the range of cancers to 20 types and further explored the associations in different genetic and lifestyle contexts. Moreover, we also evaluated the potential implications of PhenoAge in population-level cancer screening. In addition, we observed a negative association between PhenoAgeAccel and prostate cancer risk. The unexpected association may have been confounded by diabetes and altered glucose metabolism, both of which are closely linked to aging. When we removed HbA1c and serum glucose from the biological age algorithms, the association became non-statistically significant. Similar findings were also reported by *Mak et al., 2023* and *Dugué et al., 2021*.

The associations between PhenoAgeAccel and increased cancer risk may be partly attribute to a result of decline in the immune system and accumulation of environmental carcinogenic factors. Recent GWASs of PhenoAgeAccel showed that SNPs associated with PhenoAgeAccel were enriched in pathways of immune system and activation of pro-inflammatory (*Kuo et al., 2021a*; *Levine et al., 2018*). In addition to genetics, behaviors (i.e. obesity, smoking, alcohol consumption, and physical activity), and life course circumstances (i.e. socioenvironmental circumstances during childhood and adulthood) were reported to account for about 30% variances of phenotypic aging (*Liu et al., 2019*). This was in accordance with our findings that, adherence to healthy lifestyles (involving no current smoking, normal BMI, regular physical activity, and healthy diet) could slow down the aging process. In other words, these healthy lifestyles considered in our and previous studies may be causal drivers of phenotypic aging, they represent a more targetable strategy for reducing overall cancer burden by retarding the aging process. Therefore, PhenoAge provides a meaningful intermediate phenotype that can be used to guide interventions for high-risk groups and track intervention efficacy (*Liu et al., 2019*).

This study has several strengths, including a large sample size, a prospective design of the UK Biobank study, and an effective application of PhenoAgeAccel in predicting the risk of overall cancer. Nevertheless, we also acknowledge several limitations. First, we calculated PhenoAge based on 9 biomarkers from blood, which were measured at baseline. As such, we were unable to access the change of PhenoAgeAccel during the follow-up period. Second, previous studies have indicated that patricians in the UK Biobank differ from the general UK population because of low participation and healthy volunteer bias (*Fry et al., 2017*). Finally, even though the findings were achieved from participants with diverse ethnic backgrounds of the UK Biobank, the generalizability of our findings should be further assessed in more diverse populations when available.

In summary, our study showed that accelerated aging, which was measured by PhenoAgeAccel, was consistently related to an increased risk of several site-specific cancer and overall cancer with adjustment for chronological age, within and across genetic risk groups. PhenoAgeAccel can serve as a productive tool to facilitate the identification of cancer susceptible individuals, in combination

with individual's genetic background, and act as an intermediate phenotype to guide interventions for high-risk groups and track intervention efficacy.

## Acknowledgements

The authors thank the investigators and participants in UK Biobank for their contributions to this study. This work was supported by the National Natural Science Foundation of China (82230110, 82125033, 82388102 to GJ; 82273714 to MZ); and the Excellent Youth Foundation of Jiangsu Province (BK20220100 to MZ).

## Additional information

### Funding

| Funder | Grant reference number | Author |
|---|---|---|
| National Natural Science Foundation of China | 82230110 | Guangfu Jin |
| National Natural Science Foundation of China | 82125033 | Guangfu Jin |
| National Natural Science Foundation of China | 82388102 | Guangfu Jin |
| National Natural Science Foundation of China | 82273714 | Meng Zhu |
| Excellent Youth Foundation of Jiangsu Province | BK20220100 | Meng Zhu |

The funders had no role in study design, data collection and interpretation, or the decision to submit the work for publication.

### Author contributions

Lijun Bian, Data curation, Formal analysis, Validation, Visualization, Writing – original draft, Writing – review and editing; Zhimin Ma, Data curation, Formal analysis, Validation, Visualization, Writing – review and editing; Xiangjin Fu, Data curation, Validation, Visualization, Writing – review and editing; Chen Ji, Data curation, Validation, Visualization; Tianpei Wang, Formal analysis, Validation, Visualization, Methodology, Writing – review and editing; Caiwang Yan, Juncheng Dai, Methodology, Writing – review and editing; Hongxia Ma, Supervision, Project administration, Writing – review and editing; Zhibin Hu, Conceptualization, Supervision, Project administration, Writing – review and editing; Hongbing Shen, Conceptualization, Resources, Supervision, Project administration, Writing – review and editing; Lu Wang, Conceptualization, Formal analysis, Supervision, Validation, Writing – review and editing; Meng Zhu, Conceptualization, Formal analysis, Supervision, Funding acquisition, Validation, Writing – original draft, Writing – review and editing; Guangfu Jin, Conceptualization, Formal analysis, Supervision, Funding acquisition, Validation, Writing – review and editing

### Author ORCIDs

Zhibin Hu ⓘ http://orcid.org/0000-0002-8277-5234
Meng Zhu ⓘ http://orcid.org/0000-0001-5122-1733
Guangfu Jin ⓘ http://orcid.org/0000-0003-0249-5337

### Ethics

UK Biobank has received ethics approval from the Research Ethics Committee (ref. 11/NW/0382).

Reviewer #1 (Public Review): https://doi.org/10.7554/eLife.91101.3.sa1
Reviewer #2 (Public Review): https://doi.org/10.7554/eLife.91101.3.sa2
Author response https://doi.org/10.7554/eLife.91101.3.sa3

## Additional files

### Supplementary files
• Supplementary file 1. Supplementary Tables a - k for additional results. (**a**) Association results of Phenotypic Age Acceleration (PhenoAgeAccel) with site-specific cancer risk per 5 years increased. (**b**) Sensitivity analysis of association between different risk levels of PhenoAgeAccel and cancer risk. (**c**) Sensitivity analysis of association between PhenoAgeAccel and cancer risk by excluding of patients diagnosed in the first two follow-ups. (**d**) Sensitivity analysis of association between PhenoAgeAccel and cancer risk in unimputed data. (**e**) Sensitivity analysis of association between PhenoAgeAccel and cancer risk in the unrelated white British population. (**f**) Sensitivity analysis of association between PhenoAgeAccel and cancer risk using retrained PhenoAge in cancer-free participants. (**g**) Relative excess risk due to interaction (RERI) and attributable proportion (AP) for additive interaction between genetic and PhenoAgeAccel categories. (**h**) Risk of incident cancer according to PhenoAgeAccel categories within each genetic risk level. (**i**) PhenoAgeAccel of participants stratified by lifestyle factors. (**j**) Association results of PhenoAgeAccel with lifestyle factors of participants. (**k**) RERI and AP for additive interaction between genetic and lifestyle factors on PhenoAgeAccel.
• MDAR checklist

### Data availability
All data generated or analysed during this study are included in the manuscript and supporting files. The data underlying the results presented in the study are available from the UK Biobank (https://www.ukbiobank.ac.uk; Application Number: 60169).

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
