## [Editor Report · eLife assessment]

This study presents **fundamental** findings that advance our understanding of the role of phenotypic aging in cancer risk. This article presents **compelling** results that show Phenotypic Age Acceleration (PhenoAgeAccel) can predict cancer incidence of different types and could be used with genetic risk to facilitate the identification of cancer-susceptible individuals. These results will be of broad interest to the research community and clinicians.

---

## [Referee Report · Reviewer #1 (Public Review)]

Bian et al showed that biomarker-informed PhenoAgeAccel was consistently related to an increased risk of site-specific cancer and overall cancer within and across genetic risk groups. The results showed that PhenoAgeAccel and genetic liability of a bunch of cancers serve as productive tools to facilitate the identification of cancer-susceptible individuals under an additive model. People with a high genetic risk for cancer may benefit from PhenoAgeAccel-imformed interventions.

As the authors pointed out, the large sample size, the prospective design UK Biobank study, and the effective application of PhenoAgeAccel in predicting the risk of overall cancer are the major strengths of the study. Meanwhile, the CPRS seems to be a solid and comprehensive score based on incidence-weighted site-specific polygenic risk scores across 20 well-powered GWAS for cancers.

---

## [Referee Report · Reviewer #2 (Public Review)]

Bian et al. calculated Phenotypic Age Acceleration (PhenoAgeAccel) via a linear model regressing Phenotypic Age on chronological age. They examined the associations between PhenoAgeAccel and cancer incidence using 374,463 individuals from the UK Biobank and found that older PhenoAge was consistently related to an increased risk of incident cancer, even among each risk group defined by genetics.

The study is well-designed, and uses a large sample size from the UK biobank.

Comments on revised version:

The authors have addressed all my concerns.

---

## [Author Response]

The following is the authors’ response to the original reviews.

**Responses to Reviewer 1:**
It wouldn't be very surprising to identify the association between PhenoAgeAccel and cancer risk, since the PhenoAgeAccel was constructed as a predictor for mortality which attributed a lot to cancer. Although cancer is an essential mediator for the association, sensitivity analyses using cancer-free mortality may provide an additional angle.

As suggested, we retrained the PhenoAge in cancer-free participants based on mortality and recalculated PhenoAgeAccel in the UK Biobank. As expected, the re-calculated PhenoAgeAccel was still significantly associated with an increased risk of overall cancer in both men and women. The relevant results have been added to Appendix 1-table6.

It would be interesting to see, to what extent, PhenoAgeAccel could be reversed by environmental or lifestyle factors. G by E for PhenoAgeAccel might be worth a try.

As suggested, we performed interaction analysis between genetic and lifestyle factors on PhenoAgeAccel, and added the methods and results in the revision as follows:

“55 independent PhenoAgeAccel-associated SNPs (P < 5 × 10-8) and corresponding effect sizes were derived from a large-scale PhenoAgeAccel GWAS including 107,460 individuals of European ancestry (Kuo, Pilling, Liu, Atkins, & Levine, 2021). A PhenoAgeAccel PRS was created using an additive model as previously described (Dai et al., 2019). In short, the genotype dosage of each risk allele for each individual was summed after multiplying by its respective effect size of PhenoAgeAccel.” (Page 6)

“We performed additive interaction analysis between genetic risk (defined by CPRS) and PhenoAgeAccel on overall cancer risk, as well as genetic risk (defined by PhenoAgeAccel PRS) and lifestyle on PhenoAgeAccel using two indexes: the relative excess risk due to interaction (RERI) and the attributable proportion due to interaction (AP).” (Page 9)

“However, we did not observe any interaction between genetic risk and lifestyle on PhenoAgeAccel in both men and women (Appendix 1-table 11).” (Page 13)

**Responses to Reviewer 2:**
Since the UK biobank has a large sample size, it should have enough power to split the dataset into discovery and validation sets. Why did the authors use 10-fold cross-validation instead of splitting the dataset?

There may have been some misunderstandings in the interpretation of methods that 10-fold cross-validation was applied to select biomarkers when calculating PhenoAge in the previous manuscript (Levine et al., 2018). In this study, we analyzed the association between PhenoAgeAccel and incident cancer risk by dividing participants into ten groups based on the deciles of PhenoAgeAccel and assessed the associations of each group compared to the lowest decile. To avoid any confusion, we have removed the description of 10-fold cross-validation from the Methods section (Page 5).

**Recommendations for the authors:**
In addition, there is extant literature on the role of Phenotypic Age Acceleration in cancer risk and mortality that should be reviewed. Please also address possible overlap with previous work that used the UK Biobank cohort study (PMCID: PMC9958377).

As suggested, we have reviewed the association of Phenotypic Age Acceleration with cancer risk, and added it into the Discussion section as follows:

“Recently, several studies have confirmed the associations between PhenoAgeAccel and cancer risk. Mak et al. explored three measures of biological age, including PhenoAge, and assessed their associations with the incidence of overall cancer and five common cancers (breast, prostate, lung, colorectal, and melanoma) (Mak et al., 2023). In our previous study, we investigated the association between PhenoAgeAccel and lung cancer risk and analyzed the joint and interactive effects of PhenoAgeAccel and genetic factors on the risk of lung cancer (Ma et al., 2023). In comparison to these studies, our analysis expanded the range of cancers to 20 types and further explored the associations in different genetic and lifestyle contexts. Moreover, we also evaluated the potential implications of PhenoAge in population-level cancer screening.” (Page 15).

Other minor comments:Line 216, "-4.35 to -1.25" or "-4.35, -1.25" may be better.

As suggested, we have adjusted text accordingly.

Line 260, please clarify the PRS used for G by E interaction testing. It could be site-specific PRS or CPRS.

We used CPRS for G by E interaction testing, and we have changed the description of our methods as follows:

“We performed additive interaction analysis between genetic risk (defined by CPRS) and PhenoAgeAccel on overall cancer risk, as well as genetic risk (defined by PhenoAgeAccel PRS) and lifestyle on PhenoAgeAccel using two indexes: the relative excess risk due to interaction (RERI) and the attributable proportion due to interaction (AP).” (Page 9)

Line 223, The discussion/interpretation for "while negatively associated with risk of prostate cancer" is lacking.

As suggested, we have discussed this as follows:

“In addition, we observed a negative association between PhenoAgeAccel and prostate cancer risk. The unexpected association may have been confounded by diabetes and altered glucose metabolism, both of which are closely linked to aging. When we removed HbA1c and serum glucose from the biological age algorithms, the association became non-statistically significant. Similar findings were also reported by Mak et al. (Mak et al., 2023) and Dugue et al. (Dugue et al., 2021).” (Page 15).

It is not clear how to define "biologically older" and "biologically younger". Whether the individuals fall in the "middle area" will impact the results.

We defined "biologically older" and "biologically younger" based on Phenotypic Age Acceleration (PhenoAgeAccel), which was defined as the residual obtained from a linear model when regressing Phenotypic Age on chronological age. We categorized individuals with PhenoAgeAccel > 0 as biologically older and those with PhenoAgeAccel < 0 as biologically younger.

Compared with individuals at low accelerated aging (the bottom quintile of PhenoAgeAccel), we found those in the "middle area" (quintiles 2 to 4) and high accelerated aging (the top quintile) had a significantly higher risk of overall cancer (Table 2). Individuals fall in the "middle area" also had a moderate risk of overall cancer, when reclassified accelerated aging levels according to quartiles or tertiles of the PhenoAgeAccel (Appendix 1-table 2).

Do men and women have distinct biological ages, so they were analyzed separately?

We found that men (median PhenoAgeAccel: 0.34, IQR: -2.42 to 3.53) have higher biological ages than women (median PhenoAgeAccel: -1.38, IQR: -4.26 to 1.96) (P < 0.0001). In addition, men and women have different cancer incidence patterns (Rubin, 2022). Therefore, we conducted separate analyses to investigate the associations of PhenoAgeAccel with cancer risk in men and women.

Dai, J., Lv, J., Zhu, M., Wang, Y., Qin, N., Ma, H., . . . Shen, H. (2019). Identification of risk loci and a polygenic risk score for lung cancer: a large-scale prospective cohort study in Chinese populations. Lancet Respir Med, 7(10), 881-891. doi: 10.1016/S2213-2600(19)30144-4

Dugue, P. A., Bassett, J. K., Wong, E. M., Joo, J. E., Li, S., Yu, C., . . . Milne, R. L. (2021). Biological Aging Measures Based on Blood DNA Methylation and Risk of Cancer: A Prospective Study. JNCI Cancer Spectr, 5(1). doi: 10.1093/jncics/pkaa109

Kuo, C. L., Pilling, L. C., Liu, Z., Atkins, J. L., & Levine, M. E. (2021). Genetic associations for two biological age measures point to distinct aging phenotypes. Aging Cell, 20(6), e13376. doi: 10.1111/acel.13376

Levine, M. E., Lu, A. T., Quach, A., Chen, B. H., Assimes, T. L., Bandinelli, S., . . . Horvath, S. (2018). An epigenetic biomarker of aging for lifespan and healthspan. Aging (Albany NY), 10(4), 573-591. doi: 10.18632/aging.101414

Ma, Z., Zhu, C., Wang, H., Ji, M., Huang, Y., Wei, X., . . . Shen, H. (2023). Association between biological aging and lung cancer risk: Cohort study and Mendelian randomization analysis. iScience, 26(3), 106018. doi: 10.1016/j.isci.2023.106018

Mak, J. K. L., McMurran, C. E., Kuja-Halkola, R., Hall, P., Czene, K., Jylhava, J., & Hagg, S. (2023). Clinical biomarker-based biological aging and risk of cancer in the UK Biobank. Br J Cancer, 129(1), 94-103. doi: 10.1038/s41416-023-02288-w

Rubin, J. B. (2022). The spectrum of sex differences in cancer. Trends Cancer, 8(4), 303-315. doi: 10.1016/j.trecan.2022.01.013